# A Hope for Hope: Refocusing Health Promotion on Hopefulness to Reduce Alcohol Consumption and Breast Cancer

**DOI:** 10.3390/ijerph22020188

**Published:** 2025-01-29

**Authors:** Paul R. Ward, Kristen Foley, Megan Warin, Catherine Palmer, Sarah MacLean, Belinda Lunnay

**Affiliations:** 1Research Centre for Public Health, Equity and Human Flourishing, Torrens University, Adelaide, SA 5000, Australia; kristen.foley@torrens.edu.au (K.F.); catherine.palmer@torrens.edu.au (C.P.); belinda.lunnay@torrens.edu.au (B.L.); 2Fay Gale Centre for Research on Gender, School of Social Sciences, The University of Adelaide, Adelaide, SA 5005, Australia; megan.warin@adelaide.edu.au; 3Centre for Alcohol Policy Research, La Trobe University, Melbourne, VIC 3000, Australia; s.maclean@latrobe.edu.au

**Keywords:** hope, health promotion, sociology, public health, freedom

## Abstract

Our perspective paper focuses on the sociology of hope and is a call to action for health promotion policy makers to create the conditions for hopefulness in alcohol reduction policy, advocacy and programs for/with midlife women. Alcohol is a major risk factor for breast cancer, and high proportions of midlife women in most high-income countries drink at “risky” levels, increasing the chances of breast cancer (due to both age and alcohol consumption). At present, alcohol reduction approaches convey mostly individualised risk messages and imply personal responsibility for behaviour change, stripped from contexts, and heavy drinking persists among groups. New approaches that address the social norms, identities and practices that operate to sustain heavy drinking are necessary considering alcohol harms. We argue that focusing on changing these factors to support hopeful futures may create hope for midlife women to reduce alcohol consumption. We synthesise contemporary theories on the sociology of hope and analyse how these might help to refocus health promotion policy on hopefulness in the context of alcohol reduction and breast cancer prevention. We will draw on Freire’s notions of a Pedagogy of Oppression and a Pedagogy of Hope to show how enabling people to recognise and respond to the “oppressive forces” shaping their alcohol consumption might lead to more hopeful futures with reduced alcohol consumption for priority populations. Our focus on building hope into health-promoting alcohol reduction approaches intends to shift policy focus from the individual as the “problem” towards hope being a “solution”.

## 1. Introduction—Midlife Women, Alcohol Consumption and Breast Cancer Risk

Alcohol consumption remains a major societal problem globally, contributing to a myriad of health conditions. While men consume more alcohol than women, midlife women (45–64 years) experience specific risks from drinking alcohol, including increased risk of breast cancer (most common cancer in women [1]), with an estimated 10% of breast cancers caused by alcohol [2,3]. In Australia, currently, midlife women consume more alcohol than previous generations of midlife women and more than other age groups of women [4], exacerbated during COVID-19 lockdowns [5,6]. Midlife women who drink alcohol face a triple burden: (1) *all midlife women* (irrespective of alcohol consumption) are at risk of breast cancer due to their age; (2) *midlife women who drink alcohol* have an increased risk of breast cancer (a dose–response relationship: every drink throughout life accumulates breast cancer risk [7]); and (3) *midlife women in specific socio-cultural groups* face structural disadvantages which increase the likelihood of heavy alcohol consumption, further increasing breast cancer risk. Research evidence shows consistently that current health promotion policies, which rely on behaviour change, is unlikely to be effective for women who consume alcohol in response to structural disadvantages, thereby worsening future health inequities [8,9].

The key aim of this perspective paper is to call health promotion policy makers to action and create the conditions for “hopefulness” in alcohol reduction policy, advocacy and programs. Applying concepts from the sociology of hope will help public health and health promotion policy makers to understand “how/where to” change the social, economic, cultural and political conditions (oppressive forces) in order to increase hopefulness (i.e., freedom to act and choose) in particular population groups. Our paper provides a summary of the theoretical literature on hope and attempts to make it accessible to public health and health promotion audiences. Our research focuses on a contemporary health “problem” (midlife women, socio-political and commercial determinants of alcohol consumption) for which hope offers a “solution” (support hopeful futures to create hope for reduced alcohol consumption). We argue that increasing hope may lead to reduced alcohol consumption for midlife women oppressed because of social class and gender, but readers may consider if/how/why developing hope-based approaches may work in their areas of preventive health (e.g., risk behaviours like unhealthy diets, physical inactivity, having unprotected sex and smoking). Given the different ways in which hope is understood in, say, philosophy, psychology and sociology, we focus particularly on the sociology of hope since we argue that it has specific relevance to interventions and policies aimed at population-level change rather than changes in individual cognition (i.e., psychology of hope).

## 2. Historical Trajectory of Hope, Hopefulness and Links to Freedom

From Greek Philosophy, through antiquity and the Middle Ages to the present day, philosophers, sociologists, psychologists, revolutionaries and politicians agree on the centrality of hope for human flourishing [10,11,12]. Seemingly different and even polar-opposite thinkers, activists and professionals see hope as integral to what it is to be “human”. In contemporary times, the academic study of “hope” became central to understanding how and why humans managed to endure insufferable suffering, including survivors of the Holocaust [13,14,15,16]. Developing, maintaining and reinforcing hope in order to live through and past the suffering seemed to provide a “meaning” which enabled people to survive—as Freire outlines, it is the strive for freedom that is humanising [17].

Hope has been inextricably linked to freedom, with hope being the way to “see” and move towards freedom. In the context of this paper, regard freedom both as encompassing “not” being incarcerated and, more broadly, as having and exercising agency, choice and free will. Numerous social theorists argue that freedom is both a prerequisite and inextricably linked to hope [18,19,20,21]. Whilst both hope and freedom have been extensively theorised outside public health, they have been less widely used within health promotion campaigns, approaches, interventions and policies. Hope has also been linked with happiness. For example, Bauman and Ahmed both identified the links between hope and happiness, “*we are happy as long as we haven’t lost the hope of becoming happy*” [22] and “*if we hope for happiness, then we might be happy as long as we can retain this hope*” [23]. Scholars have recognised the ways happiness “inoculates against precarity” [24], which in turn allows hopeful horizons that are future-orientated. Key texts on hope, such as “The Principle of Hope” [25], “The Revolution of Hope” [20] and “The Sociology of Hope” [26] underride the consistency of hope as being something that coheres transformation, change and future acts—such as to reduce consumption of a pleasurable substance (alcohol) with the view to prevent disease (breast cancer).

## 3. A Sociology of Hope

Studies of hope (and of hopeful/lessness) encompass diverse perspectives, including philosophy, ethics, economics and sociology, in addition to self-help books, wellness literature and popular culture [27]. Hope can be diversely located within political principles, virtue ethics, emotional affect/s and social practices, being an emotion, cognitive process, existential stance, state of being, attitude, disposition, instinct, intuition and state of mind (to name a few). There are also different forms that hope can take, producing numerous typologies of hope. Desroche identified four of these: “waking dream”, “collective ideation”, “exuberant expectation” and “generalised utopia” [26]. Numerous other typologies include hopes that are failed, trapped and empty, in addition to sound hopes, false hopes, fearful hopes, transformative hopes and critical hopes [28,29,30,31]. Dichotomies have also been developed between the individual and collective, realistic and illusory, big and small hopes, and short ranges and long ranges [28]. We are not suggesting health promotion focuses on an either/or binary between “hopeful” or “hopeless”, but rather, we suggest that a sociology of hope allows for a nuanced continuum in which hopefulness is mitigated by gender, social class, age and structural factors, the intersections of which affect particular population groups’ capacity to act [32].

Even in 2013, Webb [29] identified 26 theories of hope and 54 different definitions, and in the last decade, no doubt even more have emerged and are circulating. Consistent throughout these is the situated and active nature of hopefulness. Freire, for example, articulates, “*The hoped-for is not attained by dint of raw hoping. Just to hope is to hope in vain*” [18]. The eloquent sociologist Bauman uses a metaphor from chemistry (the creation of water) to explain the interrelationship of human agency (what he terms imagination; akin to the capacity to act) and a recognition of the reality of their social circumstances (what he terms “moral sense”) on creating the conditions for hope, “*As inevitably as the meeting of oxygen and hydrogen results in water, hope is conceived whenever imagination and moral sense meet*” [19]. Bauman also said “*all of us are doomed to the life of choices…but not all of us have the means to be the choosers*” [33]. In taking both slightly further, McGeer links hope and care in what she calls “the art of good hope”, describing this as, “*both caring and careful—caring in doing one’s utmost care to create conditions under which hope thrives, but careful in understanding what those conditions are*” [30].

We use the following definition of hope to guide our argument within this paper for health promotion policy based on hopefulness: “*Hope can only arrive when you recognize that there are real options and that you have genuine choices. Hope can flourish only when you believe that what you do can make a difference, that your actions can bring a future different from the present. To have hope, then, is to acquire a belief on your ability to have some control over your circumstances. You are no longer entirely at the mercy of forces outside yourself*” [34]. This definition positions hope as being shaped by both individual agency and structural conditions, and thereby clearly of sociological interest. We take hope as a socially mediated human capacity with varying affective, cognitive and behavioural dimensions, “a *restless, future-oriented longing for that which is missing*” [35]. Whilst we recognise the role of individual agency in expressing hopes, “to *experience ourselves as agents of potential as well as agents in fact*” [30], we also recognise the structural conditions that limit “the agents of potential”, since if people believe the present or the future can only have negative outcomes, they are likely to act according to these expectations [36,37,38].

## 4. Interventions and Policies to Reduce Alcohol Consumption—Need for Hope?

There is a global gap in knowledge about socially and culturally appropriate interventions for equitably reducing alcohol consumption in midlife women. Rather than continue to problematise individual “choices” to drink, which entails expecting women to demonstrate free will and modify their behaviour, we urgently need “solutions” to the intersecting community, socio-political and commercial forces that shape alcohol consumption for midlife women [39,40]. Further to this, alcohol reduction interventions that target the individual (e.g., persuading behaviour change through screening for problematic drinking) require that consuming alcohol be recognised by the individual as a “problem.” However, the majority (87%) of Australian drinkers consider themselves “responsible drinkers”, even though 68% are risky/heavy drinkers [41], leading to alcohol reduction interventions being resisted (i.e., not identifying as problematic drinkers) [42].

We argue that new, hope-focused approaches are required to provide innovative ways to support midlife women to recognise the social, economic and commercial determinants (which we refer to as “oppressive forces” throughout this paper, in keeping with the language used by Paolo Freire [17]) that impact their alcohol consumption and the feasibility of reducing it. By developing interventions based on creating hope and/or improving hopefulness, Freire’s framework suggests we can empower women who drink heavily to resist oppressive forces which shape drinking and enable women to address these and engage in different social practices associated with reduced alcohol consumption, rather than requiring drinking to be seen as “problematic” (and resist interventions). Freire refers to the process of supporting people to recognise the oppressive forces acting upon them as raising “critical consciousness” [17,43].

If we think of hope as a vision for the future, being hopeful enables imagining that the future can be different through raising critical consciousness. It enables humans to perceive a purpose in reflecting on if/where change (such as reducing alcohol consumption to reduce risks and increase health) is possible and worthwhile. On the other hand, the absence of hope, or diminished hopefulness, creates a sense of futility in envisaging change since there is no anticipation of a future that could be different.

Recent Australian evidence suggests that attitudes to drinking alcohol are shifting [44]. Australians increasingly support action against alcohol harms, and larger proportions of people are willing to reduce alcohol consumption. This change could be a response to the unsustainability of increased drinking during COVID-19 [9], increased availability of no- and low-alcohol products (e.g., NoLos), and/or awareness of harms and increased engagement with the “sober curious” movement [45]. This evidence of social change adds impetus to the potential for new hope-focused approaches to reduce alcohol consumption for midlife women. We argue that there is huge potential to build on Freire’s concept of critical consciousness development [17,18] and co-develop health-promoting policy that enables alcohol reduction, based on Freire’s pedagogy of hope [18]. Applying this approach, midlife women will be empowered to recognise, through raising their critical consciousness, the oppressive forces currently shaping their alcohol consumption (this is based on Freire’s Pedagogy of Oppression). New approaches can then be co-developed with women and tested, which aim to either negate the impact of the oppressive forces or change them into forces for hope by raising critical consciousness to adopt new practices that resist social norms. For example, in a study about residential substance use services, researchers found that young people’s hope was located in social relationships and productive discourses and presented differently according to the external resources available to them, giving some more capacity than others to action their “hoped-for futures” [46]. This is critically important since it indicates that different groups of midlife women may require different social, cultural and/or economic resources in order to gain a sense of (or enact real) control over the oppressive forces and their imagined or “hoped-for” futures.

## 5. Utility of the Sociology of Hope for Health Promotion and Public Health

Petersen and Wilkinson [47] critique what they call individualised and privatised hopes within a hope industry, partly because hope can become a commodity which can be “traded” in a marketplace of companies selling products such as radical cures, cosmetic surgeries, diverse forms of wellness and novel health technologies. Hope is situated in differing cultural contexts too. In settler colonial Australia, some First Nation scholars embrace concepts of hope as a ballast against racism and structural violence [48], while others voice their strong opposition to waiting for the promise of hope, instead rejecting hope and elevating calls for sovereignty [49]. Most pertinent to our paper is that the structural determinants of hope can be hidden, or even invisible.

Gazing at this sociologically, McGeer [30]’s analysis of the hopes pinned on things unlikely to come to fruition—“hoping badly”—is useful and directly relates to the balance required between agency and structure. She suggests two categories of hoping badly: “wishful hope”, which is an over-reliance on external forces and systems to realise hopes, and “wilful hope”, which is an over-reliance on individual agency. Wishful hope tends to focus more on desire, “*with less focus on responsibilities of agency in both formulating and working towards the realization of one’s hopes*” (p. 113), whereas wilful hopes reflect agentic responsibilities, but wilful hopers “*have little understanding of their self-aggrandizing passions that often drive them to those ends*”, which McGeer calls “agential solipsism” (p. 116). When developing hope-based approaches, being aware of the inter-relationship of agency and structure is important to enable “the art of good hope” or “hoping well”.

The purpose of explaining “the art of good hope” is not to suggest it as a kind of self-help principal but rather to highlight a sociological way of understanding the agentic and structural antecedents and requirements for actions related to hope in line with our focus on reducing alcohol consumption for/with midlife women. Freire [17] developed a Pedagogy of Hope [18] in response to his treatise on a Pedagogy of the Oppressed, which is a similar sociological attempt to explore “how” we might support/empower the development of hope and hopeful practices in groups that recognise forms of oppression. Freire has a similar position to McGeer in terms of the requirement for agentic hopeful practices [18], arguing that meaningful, realistic and potentially achievable hopes require the development of critical consciousness, which enables a recognition of, and responses to, the structural conditions which enable/prevent hopes coming to fruition, “*I do not mean that, because I am hopeful, I attribute to this hope of mine the power to transform reality all by itself, so that I set out for the fray without taking account of concrete, material data, declaring, “My hope is enough!” No, my hope is necessary, but it is not enough. Alone, it does not win. But without it, my struggle will be weak and wobbly. We need critical hope the way a fish needs unpolluted water*” [18]. Freire’s notion of hope is similar to others who also call for critical hope [25,29,50], realistic hope [28], responsive hope [30], sound hope [51], revolutionary hope [20,52], radical hope [48] and emancipatory hope [53].

Conceptualising hope as a complex and dialectic heuristic enables its reproduction to be observed, including inequalities that shape health outcomes: like social class processes, oppressions and gendered cultural distinctions (applying this concept to our example of women, alcohol and breast cancer risk). Hoping for hope within current public health approaches and embedding hopefulness in health promotion policies means being aware of the structural conditions that enable hope and protect against hopelessness. This includes upstream approaches to addressing inequalities in women’s ability to respond to the oppressive forces by intervening with structural factors—so that they have a hopeful future which might motivate alcohol re-education [8]. Our previous research studies realise that this is currently not the case and that some women’s hopes are “dashed” before they can start thinking about alcohol reduction because of the deep oppressions located within classed (moralistic and stigmatised) and racialised social worlds [54,55]. We advocate for a gender-responsive approach to facilitating hopefulness, recognising the way that gender intersects with drivers of inequality to compromise women’s life chances—or hopes for hope [39].

## 6. Conclusions

In summary, hope has broadly been conceptualised within sociology as rational (based on evidence); irrational (based on faith); or an amalgam of both, wishful or wilful, and according to two fundamental human needs: escaping loneliness (or aiming for freedom) and the consciousness of oneself as an autonomous entity. By engaging with a sociology of hope for future health promotion policy, we are not simply trying to describe hope within different sub-population groups (based on social class, ethnicity, geographic locality, sexuality, etc.), positioning hopes as more/less enabling, but to use hope to *understand* social processes and structural inequities that impact those groups and enable or block hopefulness for a different (healthy) future. Envisage hope as the lens of a kaleidoscope—when we look through the hope kaleidoscope, we see particular social processes and their differing refractions in different social groups. Exploring the social world through the lens of hope, we consider what does hope “tell us” about the social worlds of population groups we are working with/for in public health?

In developing health promotion policy based on hopefulness, we argue that being able to recognise, critique and counteract the oppressive forces that shape “risky” practices (like heavy drinking) that characterise social worlds hinges on increasing capacities for collective critical realisations about how oppressive forces shape individual behaviour (like the ways gender inequality, socio-political settings and inadequate regulations on marketing intersect with heavy-drinking social practices). For example, a possible way of developing such hope-based interventions would be to draw on Freire’s theories and methodology to understand if/how women recognise the oppressive forces (Pedagogy of Oppression [17]) that shape their alcohol consumption practices and if/how they can develop a critical consciousness [43,56] about the oppressive forces (making them visible) and implement changes within their social worlds to disrupt and re-shape drinking practices (Pedagogy of Hope [18]). Rather than emphasise the drinking behaviours of individual women (avoiding victim blaming/stigma), this approach would foreground the interconnected gendered social practices [57] located within social worlds [58] that frame heavy-drinking for midlife women and place them at risk of alcohol harms.

## Data Availability

Not applicable.

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
