# Peer review of "A Hope for Hope: Refocusing Health Promotion on Hopefulness to Reduce Alcohol Consumption and Breast Cancer"

_ijerph, 2025, doi:10.3390/ijerph22020188_

Round 1

Reviewer 1 Report

Comments and Suggestions for Authors

In the "1. Introduction" section, you give a nice paragraph about midlife women, alcohol, and breast cancer risk, then you go straight into how the key aim of this paper is to give summary of literature about hope.  It is a huge leap and it would be nice if the 1st paragraph is followed by at least 2-4 sentences (in paragraph) to introduce 'hope'.  OR, maybe rearrange the 2nd paragraph so that you discuss 'hope' before mentioning the key aim of the paper.

Otherwise, the paper is well-written. 

Your concept of hopefulness is interesting.  I hope you will publish more about this topic in the future.

Reviewer 2 Report

Comments and Suggestions for Authors

 read the manuscript from a public health perspective. The topic is interesting and addressing it in middle aged women is necessary. But, I expected an indeep integration of the theory on sociology of hope in the discussion of alcohol consumption prevention and breast cancer. 

Reviewer 3 Report

Comments and Suggestions for Authors
  1. The authors present a narrative which aims to investigate the potential benefits of refocusing health promotion on hopefulness for reducing alcohol intake by midlife women and, thus, decreasing alcohol consumption related breast cancers. 
  2. The notion of health promotion being based on pedagogy of hope is a novel one and this notion can be easily adopted for women who seek support for reducing their alcohol intake. This field addresses a specific gap in the discipline of oncology as it specifically targets the improvement in the protection of alcohol consuming women from breast cancer. 
  3. There are a number of studies which focus on the hazardous effects of alcohol comnsumption on possible carcinogenesis in midlife women However, this study has distinct merit as it evaluates the value of the pedagogy of hope for women who wish or have will to attenuate their alcohol intake for the sake of protection form breast cancer.
  4. There is no methodology of the study as it is a narrative based on the findings of previouly published perspective studies and clinical trials.
  5. The conclusions achieved by this study are in accordance with the evidence and arguments presented and these conclusions address the main question that has been displayed.
  6. The majority of the references are appropriate, relevant and up-to-date. I would recommend that the references that were published before 2010 should be replaced with newer and more up-to-date ones if possible. 
  7. The authors should mention the limiations of their narrative based on previous perspective studies and clinical trials and they should also discuss about the clinical implications of their findings (i.e. providing psychological support for women who are at risk for breast cancer)
  8. I would recommend that the revised version of this manuscript can be reviewed once more after required corrections have been made. 
